# Antioxidant and Anti-Inflammatory Properties of Quail Yolk Oil via Upregulation of Superoxide Dismutase 1 and Catalase Genes and Downregulation of EIGER and Unpaired 2 Genes in a *D. melanogaster* Model

**DOI:** 10.3390/antiox13010075

**Published:** 2024-01-05

**Authors:** Muhammad Sani Ismaila, Kamaldeen Olalekan Sanusi, Uwaisu Iliyasu, Mustapha Umar Imam, Karla Georges, Venkatesan Sundaram, Kegan Romelle Jones

**Affiliations:** 1Department of Basic Veterinary Sciences, School of Veterinary Medicine, The University of the West Indies, St. Augustine 999183, Trinidad and Tobago; sani.muhammad@sta.uwi.edu (M.S.I.); karla.georges@sta.uwi.edu (K.G.); venkatesan.sundaram@sta.uwi.edu (V.S.); 2Centre for Advanced Medical Research and Training (CAMRET), Usmanu Danfodiyo University, Sokoto 840004, Nigeria; sanusikamaldeen@yahoo.com (K.O.S.); mustyimam@gmail.com (M.U.I.); 3Department of Pharmacognosy and Drug Development, Faculty of Pharmaceutical Sciences, Kaduna State University, Kaduna 800283, Nigeria; uwaisunka@gmail.com

**Keywords:** antioxidant effects, *EIGER* and *UPD2* gene down-regulation, fruit fly, FTIR, GC–MS, quail egg oil extracts

## Abstract

Quail egg yolk oil (QEYO) has a rich history of medicinal use, showcasing heightened antioxidant and bioactive properties in our prior studies. This positions QEYO as a promising candidate for therapeutic and cosmetic applications. In this investigation, QEYO was extracted using ethanol/chloroform and 2-propanol/hexane solvents. GC–MS and FTIR analyses quantified 14 major bioactive compounds in the ethanol/chloroform fraction and 12 in the 2-propanol/hexane fraction. Toxicity evaluations in fruit flies, spanning acute, sub chronic, and chronic exposures, revealed no adverse effects. Negative geotaxis assays assessed locomotor activity, while biochemical assays using fly hemolymph gauged antioxidant responses. Real-time PCR revealed the relative expression levels of the antioxidant and anti-inflammatory genes. FTIR spectra indicated diverse functional groups, and the GC–MS results associated bioactive compounds with the regulation of the anti-inflammatory genes *EIGER* and *UPD2*. While no significant change in SOD activities was noted, male flies treated with specific QEYO doses exhibited increased catalase activity and total antioxidant capacity, coupled with a significant decrease in their malondialdehyde levels. This study offers valuable insights into the bioactive compounds of QEYO and their potential regulatory roles in gene expression.

## 1. Introduction

Quails and their products, especially eggs, have long been recognized for their medicinal potential [1,2]. Quail meat and eggs are gaining popularity worldwide as awareness of their medicinal benefits has increased [3,4,5]. It has been discovered that quail meat is becoming a popular delicacy in Europe, where around 9 million wild quails are harvested annually from Turkey, Portugal, and Spain and transported to France, where over 20 million tons of quail meat is processed annually for human consumption [6]. The Japanese quail egg has been reported to contain nutrients four times the value of chicken eggs [7] and contain a higher amount of crude protein, fat, and mineral ash per unit egg weight than the traditionally consumed chicken egg and other poultry species. Functional proteins such as ovomucoid, ovotransferrin, and lysozymes have also been documented [8,9].

Oils from natural products have widely been known for their use as medicines, excipients, or carriers of various medicinal substances used in the treatment of various ailments. Pisseri et al. [10] reported that essential oils contain a mixture of compounds that need to be properly quantified to define the standards for their safety and efficacy. These essential compounds have various medicinal and therapeutic activities, ranging from spasmolytic, revulsive, anti-inflammatory, decongestant, immunomodulatory, antimicrobial, antimitotic, expectorant, mucolytic, antioxidant, psychotropic, analgesic, and acaricidal activities. Chicken egg yolk oil (CEYO) has demonstrated anti-inflammatory and wound-healing activities [5,11]. The analgesic and anti-inflammatory effects of duck and chicken egg yolk have been previously reported [12,13]. This anti-inflammatory effect was postulated to regulate the Nrf2/NF-κB pathway in Caco3 cell lines in vitro [14]. It has been documented that quail eggs have better cosmetic effects than chicken eggs due to their high contents of egg yolk, dry matter, and vitamin A, and have higher stability, making them more resistant to decomposition [15]. Our previous study showed that quail egg yolk oil contains more antioxidants and other chemical components such as saponins, anthraquinones, and other bioactive substances, which are higher compared to other products [16]. Hen egg is known for its hemostatic effect in the treatment of acute and chronic eczema [12,17,18].

In this study, our aim was to analyze the different compounds in QEYO via both GC–MS and FTIR, investigate their antioxidant activity, and quantify their toxicity. A further mechanistic study will also reveal the effects of QEYO on the expression of antioxidant and anti-inflammatory genes in the fruit fly model.

## 2. Materials and Methods

### 2.1. Source and Extraction of Oil from Liquid Quail Yolk

The quail eggs were procured from a local vendor in St. Augustine, Trinidad and Tobago. The oil extraction from liquid egg yolk involved a combination of two solvents, as previously outlined [19]. Specifically, a solvent mixture of ethanol/chloroform (30/70, *v*/*v*) was used for the first extraction group, while 2-propanol/hexane (30/70, *v*/*v*) was employed for the other extraction group.

### 2.2. Fourier-Transform Infrared Spectroscopy (FTIR) Analysis

The FTIR analysis was conducted utilizing the Agilent Cary 630 FTIR spectrometer (Agilent Technologies, Santa Clara, CA, USA) with a base system featuring high emissions of infrared radiation. This base system operates in conjunction with a 5-bounce diamond ATR sampling accessory equipped with an internal reflection element (IRE) crystal. This analysis was executed in transmission mode, and the signal was recorded within the wavelength range of 4000–650 cm^−1^. The sample spectra were generated using MicroLab Software version 5.6 (Agilent Technologies, Santa Clara, CA, USA), as reported previously [20].

### 2.3. Gas Chromatography/Mass Spectrometry (GC–MS) of QEYO Bioactives

The GC–MS analysis was conducted using the Agilent Technologies Intuvo 9000 GC system and the Agilent Technologies 5977B mass selective detector (MSD) coupled with the 4513A automatic liquid sampler (ALS) (Agilent Technologies, Santa Clara, CA, USA). The column that was used had the part number Agilent 1909IS-483UI-INT capillary column with the specification HP-5MS UI 30 m, 0.25 mm, and 0.25 µm (Intuvo, Scotts Valley, CA, USA). Helium served as the carrier gas with a flow rate of 1.2 mL/min. The injection volume was 1 µL. The inlet temperature was maintained at 300 °C. The oven temperature was initially programmed at 50 °C for 5 min at a rate of 5 °C. A total run time of 58 min was employed. The MSD transfer line was held at a temperature of 250 °C. The source temperature was 230 °C, and the MS quad was set at 150 °C. Electron ionization at 70 eV was employed as the ionization mode. The total ion count (TIC) was utilized for the identification and quantification of compounds. The spectrum of the separated compound was compared with the database of spectra of known compounds stored in the NIST05 reference spectra library. Data analyses and peak area measurement were carried out using Agilent Mass Hunter software version 12.0 (Agilent Technologies, Santa Clara, CA, USA) [21].

### 2.4. Fly Culture

The fruit flies that were utilized in this study belonged to the *Drosophila melanogaster* w1118 strain, obtained from the Drosophila laboratory at the Centre for Advanced Medical Research and Training (CAMRET), Usmanu Danfodiyo University, Sokoto, Nigeria. These flies were kept on a standard cornmeal diet comprising white corn flour, yeast, agar, methylparaben, and distilled water. They were cultured in 50 mL of tube media at room temperature (ranging from 22 to 25 °C) with a relative humidity of 50–60%. All the flies that were included in this study were aged between 1 day old and 3 days old, unless explicitly mentioned otherwise.

### 2.5. Experimental Design

The flies were categorized into three study groups (groups A, B, and C) based on the extraction solvent. Group A comprised flies that were fed a diet supplemented with QEYO, extracted using a combination of ethanol and chloroform solvents. Group B included the flies receiving a diet supplemented with QEYO, extracted using a combination of 2-propanol and hexane solvents. Group C, designated as the control group, consisted of male and female flies on a standard diet with no supplementation. Within groups A and B, further divisions were made into eight subgroups based on sex and the dose of the added extract. Figure 1 illustrates these group divisions. Each treatment group utilized two media vials, with 20 flies per vial.

### 2.6. Toxicity Studies

The extracts underwent testing for acute (24 h), sub-chronic (7 days), and chronic (4 weeks) toxicity in fruit flies, conducted in two batches. In the first batch, doses of 100 mg, 50 mg, 25 mg, and 12.5 mg per gram of the standard diet were tested. The second batch involved doses of 10 mg, 5 mg, 2.5 mg, and 1.25 mg per gram of the prepared standard diet. The flies were transferred into fresh medium after each three-day period.

### 2.7. The Negative Geotaxis Assay

The negative geotaxis ability was assessed to evaluate locomotor activity following the conclusion of the treatment period before sacrifice. Ten flies per treatment group were briefly anesthetized on ice and then transferred into empty 50-mL graduated cylinders. The cylinder was marked 6 cm from the bottom. The flies were given 10 min to acclimate at room temperature. Subsequently, the cylinder was gently tapped to ensure that all the flies were at the bottom of the tube, and the count of flies passing the 6 cm mark within ten seconds was recorded. This process was repeated three times for each treatment group, with a 2-min break between repetitions.

### 2.8. The Biochemical Assay

Ten flies per treatment group were anesthetized on ice and promptly placed in a dish, followed by rinsing with 100 µL of cold PBS. Subsequently, the flies were transferred into a 1.5-mL tube, and 200 µL of cold PBS was added before rapid homogenization using a pellet pestle on ice. The homogenized samples were then centrifuged at 14,000 rpm for 3 min in a refrigerated centrifuge. The resulting supernatant containing the hemolymph was carefully collected into a new 1.5-mL tube for subsequent biochemical analyses [22].

### 2.9. The Superoxide Dismutase (SOD) Assay

The SOD level in the hemolymph was determined using a colorimetric assay kit from Solarbio Life Sciences, Beijing, China. Its principle is based on the dismutation of superoxide anions into molecular oxygen or H_2_O_2_. The superoxide anion, generated via the xanthine and xanthine oxidase reaction system, produced a blue formazan, and the absorbance was detected at 560 nm. The reagents were added to the sample, control, and blank samples 1 and 2, mixed, and incubated at room temperature for 30 min. Subsequently, the absorbance was measured at 560 nm, and the level of SOD activity was calculated according to the manufacturer’s protocol.

### 2.10. The Total Antioxidant Capacity (TAOC) Assay

The total antioxidant and antioxidant enzyme contents of hemolymph were determined using a colorimetric kit from Solarbio Life Sciences, Beijing, China. The reaction principle involves the reduction of Fe^3+^-TPTZ to blue Fe^2+^-TPTZ. The reaction mixture was prepared, the absorbance was read at 593 nm, and the total antioxidant capacity was then calculated according to the manufacturer’s protocol.

### 2.11. The Malondialdehyde (MDA) Assay

The *MDA* levels in the hemolymph were determined using a lipid peroxidation assay kit from Solarbio Life Sciences, Beijing, China. This reaction generates *MDA* and thiobarbituric acid (TBA) under acidic conditions and high temperatures. Briefly, 60 µL of the *MDA* working reagent was added to the sample tube and blank, followed by the addition of 20 µL of the sample to the sample well alone. Then, 20 µL of double-distilled water was added to the blank tube alone, and 20 µL of reagent III was added to both tubes. The mixtures were incubated at 100 °C for 1 h and cooled in an ice bath. Subsequently, they were centrifuged at 10,000× *g* for 10 min under room temperature. The supernatant was collected for absorbance measurements. The difference in absorbance at 532 nm, 450 nm, and 600 nm was used to calculate the level of *MDA* using the following formula:$$MDA\;(nmol/g)=\frac{5\;(6.45\;\times \left(\Delta A532-\Delta A600\right)-\;1.29\times \;\Delta A450)}{sample\;weight}$$

### 2.12. The Catalase (CAT) Assay

The *CAT* levels in the hemolymph were determined using a colorimetric kit from Solarbio Life Sciences, Beijing, China. Its principle is based on the action of *CAT* in decomposing H_2_O_2_ into H_2_O and O_2_. The sample was mixed with a preheated *CAT* working solution according to the manufacturer’s protocol, and the absorbance was measured at 240 nm. The activities of *CAT* were calculated based on the rate of change in absorbance according to the following formula:$$CAT\;activity\;(U/mL)\;=\frac{\left(\Delta A\times Extraction\;volume\right)}{(Ɛ\;\times \;d)\;\times \;10^{9}}÷\frac{Sample\;Volume\;}{Reaction\;time}$$
where *ε*—molar coefficient, and *d*—light path. 

### 2.13. Gene Expression Study

Based on the biochemical changes exhibited by the high dose of the extracts, the mRNA levels of the flies that received the high dose were subsequently quantified to evaluate their mechanistic basis.

#### 2.13.1. RNA Extraction

Ten fruit flies from each group were anesthetized on ice and transferred on ice into a 1.5-mL centrifuge tube. RNA was extracted using the spin column-based nucleic acid isolation kit (Daan Gene Co., Ltd., SunYat-sen University, Guangdong, China), according to the manufacturer’s instructions. In brief, the flies were homogenized on ice in lysis buffer using a micro pestle. Chloroform was added to the mixture, vortexed, and centrifuged at 12,000× *g* for 3 min at 40 °C in a refrigerated centrifuge (MX-301 Highspeed, Tomy Kogyo Co., Ltd., Tagara, Japan). The aqueous supernatant was withdrawn into a new tube and precipitated with absolute ethanol. This mixture was passed through the spin column after incubation at 720 °C for 10 min. The column was spun at 12,000× *g* for 1 min at 40 °C, and the flow-through was discarded. The pellet was washed with a deionized solution and inhibitor remover at 12,000× *g* for 1 min at 40 °C. Elution was performed with 50 µL of an eluent. RNA quality was checked using a BioSpec-nano spectrophotometer (Shimadzu Biotech, Kawasaki, Japan). RNA was considered acceptable at purities of 1.8 and 2.0 for A260/230 and A260/280, respectively.

#### 2.13.2. Primer Design

Primers for *SOD1, CAT, EIGER, UPD2*, and a housekeeping gene (RPL-32) were designed using the PrimerQuest qPCR assay design tool from Integrated DNA Technologies (IDT) after obtaining the FASTA format of the mRNA sequences from the GenBank database (http://www.ncbi.nlm.nih.gov/nucleotide/) accessed on 3 November 2020. The primer sequences have been listed in the following table (Table 1).

#### 2.13.3. Quantitative Real-Time PCR Analysis

Quantitative reverse transcription polymerase chain reaction (qRT-PCR) was conducted using a one-step qRT-PCR Sybr reagent (Toroivd Technology Company Ltd., Shanghai, China) with the Rotor-Gene Q thermal cycler (Qiagen, Hilden, Germany). The reaction mixture was prepared by combining RNA samples (2 µL each), 10 µL of qRT-PCR master mix, 1 µL of manganese (Mn), 0.4 µL of forward/reverse primer, and 6.2 µL of DNAse/RNase-free water (PCR-grade water). This mixture was gently vortexed and loaded into the thermal cycler set to the following cycling conditions: reverse transcription at 95 °C for 12 min, followed by 40 cycles of denaturation at 95 °C for 15 s, annealing at 65 °C for 30 s, and extension at 72 °C for 30 s. The fold change of each gene was determined using the comparative CT method (2^−ΔΔCT^).

### 2.14. Statistical Analysis

All statistical analyses were conducted using SPSS version 20.0 (IBM, Armonk, NY, USA). All data were expressed as the mean ± SEM from three independent experiments (n = 3). A test for normality on the data was performed, and the data were found to be normally distributed. The data were analyzed using the one-way ANOVA statistical test followed by the Tukey–Kramer post-hoc test to assess for the difference between the means of the treated groups, with a *p*-value of <0.05 considered significant. The gene expression data were also normally distributed and were analyzed via the one-way ANOVA in JMP Statistical Software Version 17.2 (SAS Institute Inc., Cary, NC, USA).

## 3. Results

### 3.1. Quail Egg Oil (QEYO) Yield

The percentage yield for the fraction of quail egg extracted using the ethanol/chloroform solvent was 25.7 ± 1.32%, while the percentage yield for the quail egg yolk extracted using the 2-propanol/hexane combination was 29.04 ± 1.72%.

### 3.2. FTIR Analysis

The FTIR bands in the QEYO fraction extracted with both the ethanol/chloroform and the 2-propanol/hexane combination showed different functional groups, indicating different compounds.

#### 3.2.1. Ethanol/Chloroform QEYO Extract

The observed absorption band in the range of 2855–2922 cm^−1^ indicated C-H stretching, a typical vibrational pattern of a methyl group showing a fatty acid chain. The band that was observed at 1742 cm^−1^ (C=O stretching) indicated a saturated aldehyde. Other bands included C-O stretching at 1381–1463 cm^−1^, P=O stretching (1237 cm^−1^), S=O stretching (1162 cm^−1^), C-N stretching (1094 cm^−1^), and Si-H deformation at 805 cm^−1^ (Figure 2A).

##### 3.2.2. 2-Propanol/Hexane Quail Egg Yolk Oil Extract

The bands that were obtained in the 2-propanol/hexane fraction of the extract were C-H stretching (2847–2922 cm^−1^), C=O stretching at 1784 cm^−1^, S=O stretching (1459.3 cm^−1^), C-O stretching at 1377 cm^−1^, Si methyl group vibration (1233 cm^−1^), C-N stretching (1094 cm^−1^), and Si-H deformation at 715–818 cm^−1^ (Figure 2B).

### 3.3. GC–MS Analysis

#### 3.3.1. Ethanol/Chloroform Quail Egg Yolk Oil Extract

GC–MS analysis of the QEYO fraction extracted with ethanol and chloroform revealed eight major compounds, with cholesterol showing the highest percentage (41.64%), followed by cis-13-octadecenoic acid, methyl ester (15.01%), hexadecanoic acid, methyl ester (10.58%), methyl stearate (7.42%), methyl 10-trans,12-cis-octadecadienoate (5.67%), arachidonic acid (2.51%), 9-octadecenoic acid (Z)-, 2-hydroxyethyl ester (1.22%), and desmosterol (1.04%) (Figure 3A). Other important compounds that were extracted include doconexent (0.82%) and diltiazem (0.46%).

##### 3.3.2. 2-Propanol/Hexane Quail Egg Yolk Oil Extract

The propanol/hexane GC–MS results showed six major compounds with cholesterol (50.32%); cholesta-3,5-diene (7.03%); cholesta-4,6-diene-3-ol (5.36%); ethylbenzene (4.22%); p-xylene (3.24%); cholesta-4,6-diene-3-ol (3. beta.) (2.94%); and benzene, 1,3-dimethyl (2.80%) (Figure 3B). 

### 3.4. Toxicity Study

The flies were transferred to the control diet and different doses of the interventional diet. They were then observed for 24 h.

#### 3.4.1. Acute Toxicity

The flies were observed for 24 h, and no mortality was observed across all the treatment and control groups.

#### 3.4.2. Sub-Chronic Toxicity

The flies were observed for 7 days, and no mortality was observed across all the treatment and control groups.

#### 3.4.3. Chronic Toxicity

The flies were observed for 28 days, and no mortality was observed across all the treatment and control groups.

#### 3.4.4. The Negative Geotaxis Assay

There were no significant differences in the locomotor activities of male and female flies treated with low (1.25 mg), mild (2.5 mg), moderate (5 mg), and high (10 mg) doses of ethanol/chloroform and 2-propanol/methanol extracts compared to the control group after 28 days of exposure (Figure 4).

### 3.5. Biochemical Assays

#### 3.5.1. Superoxide Dismutase (SOD) Activity

No significant differences were observed in the SOD activities across all the treatment groups compared to the control group in both male and female flies (Figure 5).

#### 3.5.2. *Catalase* Activity

In male flies, the treatment with 10 mg of ethanol/chloroform extract and 5 mg and 10 mg of 2-propanol/hexane extract resulted in a significant increase in catalase activities. Conversely, in female flies, the ethanol/chloroform extract, at doses of 5 mg and 10 mg, led to increased catalase activities, whereas no significant difference was observed in the flies treated with the 2-propanol/hexane extract compared to the control (Figure 6).

#### 3.5.3. Total Antioxidant Capacity

Male flies treated with all doses of the 2-propanol/hexane extract exhibited a significant increase in total antioxidant capacity, while no significant difference was observed in the flies treated with the ethanol/chloroform extract. In female flies, both the ethanol/chloroform and 2-propanol/hexane extracts significantly increased total antioxidant capacity at all doses, except for the low (1.25 mg) dose of 2-propanol/hexane compared to the control (Figure 7).

#### 3.5.4. Malondialdehyde Levels

In male flies, the administration of mild (2.5 mg), moderate (5 mg), and high (10 mg) doses of the ethanol/chloroform extract, as well as mild and high doses of the 2-propanol/hexane extract, led to a significant reduction in malondialdehyde (*MDA*) levels. Similarly, in female flies, all doses of the ethanol/chloroform extract, along with moderate and high doses of the 2-propanol/hexane extract, exhibited a significant decrease in *MDA* levels compared to the control (Figure 8).

### 3.6. Gene Expression Study

A total of four genes were analyzed, including two antioxidant genes *(CAT* and *SOD1*) and two inflammatory genes (*UPD2* and *EIGER*). No significant difference was observed in the expression of the *CAT* gene in the treated groups (*p* > 0.05) when compared to the control non-treated group (Figure 9). The *SOD1* gene showed an increase in its level of expression, although not statistically significant (*p* > 0.05), in the flies in the group treated with extract B (EXT_B) when compared to the group treated with extract A and the control group (Figure 9).

The *EIGER* gene was significantly downregulated (*p* < 0.0001) in the flies treated with the QEYO extracted with the ethanol/chloroform combination compared to the control group. A significant decrease was also observed in the expression of this gene in female flies with the QEYO extracted using both the 2-propanol/hexane and ethanol/chloroform combinations (*p* < 0.05) (Figure 9). Furthermore, a significant decrease was observed in the expression of the *UPD2* gene in the flies treated with the two QEYO extracts compared to the control non-treated group (*p* < 0.0002) (Figure 9).

When considering the sexes, a significant (*p* < 0.05) difference was observed in the *CAT* gene of both sexes. It was noted that females expressed the *CAT* genes more than their male counterparts (Figure 10).

## 4. Discussion

Natural food ingredients are considered richer and safer sources of antioxidants compared to synthetic sources, and edible oils have been reported to contain high levels of antioxidants [23]. Additionally, oils have been utilized in the treatment of diseases [23,24,25]. Our findings suggest the presence of various bioactive compounds with antioxidant and anti-inflammatory effects in the QEYO. The bands in the FTIR spectra (Figure 2) indicate various essential fatty acids involved in the modulation of metabolic and non-metabolic diseases, such as diabetes, atopic eczema, cancer, inflammatory diseases, and cardiovascular diseases [26,27,28,29].

The GC–MS spectra obtained from the chloroform/hexane fraction of the QEYO revealed about eight compounds with anti-inflammatory effects, while three compounds were obtained in the 2-propanol/hexane fraction (Table 2 and Table 3). Notably, various octadecanoic acids (OOAs) were extracted, such as methyl 10-trans, 12-cis-octadecadienoate, cis-13-octadecenoic acid, methyl ester, and methyl stearate, which are monounsaturated fatty acids with antimicrobial, antioxidant, anticancer, and anti-aging properties (Table 2). OOA inhibits LPS-induced nuclear factor (NF)-κB signaling by reducing the phosphorylation of the IκB-α and p50 proteins, significantly reducing pro-inflammatory signaling and resulting in a reduction in the levels of pro-inflammatory mediators [30].

Arachidonic acid (ARA), one of the major n-6 LCPUFAs, exerts its well-studied anti-inflammatory effect by blocking prostaglandin synthesis through the lipoxygenase and cyclooxygenase pathways [31,32]. Desmosterol has been reported to suppress macrophage inflammation activation and protect against vascular inflammation and atherosclerosis (Table 2) [33]. Cholesta-3,5-diene promotes wound healing by enhancing fibroblast migration, increasing extracellular matrix production and capillary formation, and recruiting immune cells. Its cytotoxic effect has also been documented (Table 3).

Our previous study demonstrated that oils extracted from quail egg yolk exhibit higher antioxidant levels and other bioactive compounds than those extracted from some plant sources. The propanol/hexane fraction contains p-xylene, which has been shown to have antipsoriatic, antimicrobial, antioxidant, and antifungal activity (Table 3). In another study, Lee et al. [34] reported the synthesis of ibuprofen from p-xylene, which may indicate the analgesic and anti-inflammatory activity of the compound extracted from quail egg oil.

Doconexent, an omega-3 fatty acid identified at 0.82% in the present study, is a high-docosahexaenoic acid (DHA) supplement with anti-inflammatory effects. It acts as a ligand at PPARs that has an anti-inflammatory effect and regulates inflammatory gene expression and NF-κB activation [35] (Table 2). It serves as the primary structural component of the human brain, cerebral cortex, skin, and retina, playing an important role in their functions [35].

**Table 2 antioxidants-13-00075-t002:** The GC–MS peaks, retention time, area, name, and molecular formula of the compounds in QEYO samples that were extracted using the ethanol/chloroform solvent combination.

Peak	RT (min)	Area	Compound Name	Molecular Formula	Pharmacological Use of the Most Important Compounds
20	35.082	10.58	Hexadecanoic acid, methyl ester	C_17_H_34_O_2_	Antibacterial [36], antifungal [37], anti-inflammation, antifibrosis, and peripheral vasodilation effects [38]
23	35.717	0.68	n-Hexadecanoic acid	C_16_H_32_O_2_	Anti-inflammatory [39], antibacterial, and antioxidant activities [40]
24	36.335	0.32	1-Tetracosene	C_24_H_48_	Cytotoxic effect [41]
25	37.926	0.09	Gamolenic acid	C_18_H_30_O_2_	Anti-inflammatory, antithrombotic, antiproliferative, and lipid-lowering effects [42]
26	38.223	5.67	Methyl 10-trans,12-cis-octadecadienoate	C_19_H_34_O_2_	Antifungal and antioxidant [43]
27	38.349	15.01	cis-13-Octadecenoic acid, methyl ester	C_19_H_36_O_2_	Anti-inflammatory [44]
29	38.796	7.42	Methyl stearate	C_19_H_38_O_2_	Antibacterial, antioxidant, antifungal [45], antidiarrheal, cytotoxic, and anti-inflammatory [46]
30	38.979	0.93	Oleic Acid	C_18_H_34_O_2_	Decreases myocardial infarction rate, platelet aggregation and secretion of TXA2, reduces systolic blood pressure, and improves immunity [47]
37	41.090	2.51	Arachidonic acid	C_20_H_32_O_2_	An integral constituent of the biological cell membrane that is necessary for the function of all cells, especially in the nervous system, skeletal muscle, and immune system. It modulates the function of ion channels, several receptors, and enzymes via activation as well as inhibition [48]
45	44.226	0.82	Doconexent	C_22_H_32_O_2_	DHA acts as a ligand at PPARs that has an anti-inflammatory effect and regulates inflammatory gene expression and NFκB activation [35]
46	44.306	0.46	Diltiazem	C_22_H_26_N_2_O_4_S	A calcium channel blocker that is clinically used as an antihypertensive, anti-arrhythmic, and anti-anginal agent for the management of cardiovascular conditions, such as hypertension, chronic stable angina, atrial fibrillation, and atrial flutter [49]
55	47.751	1.22	9-Octadecenoic acid (Z)-, 2-hydroxyethyl ester	C_20_H_38_O_3_	Antimicrobial and antifungal [50]
60	53.639	41.64	Cholesterol	C_27_H_46_O	Helps build new tissue and repairs damage to existing tissue, produces steroid hormones, including estrogen, helps create bile in the liver, and aids in the production of vitamin D [51]
61	54.125	1.04	Desmosterol	C_27_H_44_O	Suppresses macrophage inflammasome activation and protects against vascular inflammation and atherosclerosis [33]

**Table 3 antioxidants-13-00075-t003:** The GC–MS peaks, retention time, area, name, and molecular formula of the compounds in QEYO samples that were extracted using a 2-propanol/hexane solvent combination.

Peak	RT	Area	Compound	Molecular Formula	Pharmacological Use
6	7.216	3.24	p-xylene	C_8_H_10_	Antipsoriatic, antimicrobial, antioxidant, and antifungal activity [52,53]
8	7.994	2.80	Benzene, 1,3-dimethyl	C_8_H_10_	Anti-tumor [54]
38	14.975	0.77	Undecane	C_11_H_24_	Antiallergic and anti-inflammatory [55]
87	43.791	0.47	Oleic acid, 3-hydroxypropyl ester	C_21_H_34_O_3_	Protection against cardiovascular diseases [56]
88	44.186	0.12	Octadecanoic acid, 2,3-dihydroxypropyl ester	C_21_H_42_O_4_	Antimicrobial and anticancer effects [57]
91	49.679	0.30	Squalene	C_30_H_50_	In animals, supplementation of the diet with squalene can reduce cholesterol and triglyceride levels. In humans, it potentiates the effects of some cholesterol-lowering drugs. It functions in the skin as a quencher of singlet oxygen, protecting the skin from lipid peroxidation due to exposure to UV and other radiations. Its primary therapeutic use is as an adjunctive therapy in a variety of cancers [58]
94	50.366	2.94	Cholesta-4,6-dien-3-ol, (3.beta.)	C_27_H_44_O	Anticancer effect [59]
95	50.646	7.03	Cholesta-3,5-diene	C_27_H_44_	Promotes wound healing [60] and exhibits a cytotoxic effect [52]
96	52.557	0.27	Silicic acid, diethyl bis (trimethylsilyl) ester	C_10_H_28_O_4_Si_3_	Anti-inflammatory, antibacterial, and antioxidant effects [61]
97	53.559	50.32	Cholesterol	C_27_H_46_O	Helps build new tissue and repairs damage to existing tissue, produces steroid hormones, including estrogen, helps create bile in the liver, and aids in the production of vitamin D [51]
98	54.091	0.90	4-Dehydroxy-N-(4,5-methylenedioxy-2-nitrobenzylidene) tyramine	C_16_H_14_N_2_O_4_	Antibacterial, antidepressant, antidiarrheal, and anti-inflammatory effects [62]
99	55.453	1.86	Cholest-4-en-3-one	C_27_H_44_O	A novel drug candidate for amyotrophic lateral sclerosis [63]. An oxime with neuroprotective and antinociceptive activity [64]. It also has an anti-obesity effect [65]

Diltiazem, extracted at 0.46%, is a well-known calcium channel blocker used in the management of hypertension and angina [66]. Squalene, a free radical scavenger, has an anticarcinogenic effect, especially when combined with oleic acid, and is found in many edible oils, such as olive and palm oil [67]. Silicic acid was also obtained from the 2-propanol/hexane fraction. This antioxidant compound is a powerful radical scavenger that has been reported to have a strong free radical quenching effect in several essential oils and other bioactive products [68,69,70]. It was also reported to have an anti-inflammatory effect (Table 2).

Other compounds that were reported to have anti-inflammatory effects in the propanol/hexane fraction of the QEYO extract include undecane (0.77%), which was recently reported to have antiallergic and anti-inflammatory effects, especially in inflammatory conditions related to the skin, where it increases intracellular cAMP levels in mast cells and keratinocytes and inhibits degranulation and the secretion of histamine and tumor necrosis factor α (TNF-α) in sensitized mast cells. It reversed p38 phosphorylation and nuclear factor kappa B (NF-κB) transcriptional activity, and targeted cytokine/chemokine genes [44]. 4-Dehydroxy-N-(4,5-methylenedioxy-2-nitrobenzylidene) tyramine (0.90%) interacts with monoamine oxidase, serotonin, in its effects on depression and diarrhea [71].

It has been observed in this study that cholesterol has the highest peak in both fractions with 41.64% and 50.32%, respectively. The Hellenic National Nutrition and Health Survey (HNNHS) reported that eggs, a main source of dietary cholesterol, do not increase the risk of dyslipidemia; in contrast, they indicated that the risk of dyslipidemia could decrease with moderate egg consumption [72]. Cholesterol and its metabolites influence bone homeostasis by modulating the differentiation and activation of osteoblasts and osteoclasts [73,74,75].

The toxicity study with fruit flies confirmed the safety of our extracts in acute and chronic toxicity tests, including the negative geotaxis assay. The flies that were treated with QEYO exhibited an increase in their total antioxidant and catalase levels. The gene expression mechanism provides a clear picture of the effects of our extraction, stimulating antioxidant genes and downregulating anti-inflammatory genes.

The mRNA expression levels of the *SOD1* antioxidant genes increased in the treated group compared to the control group, although not significantly. Meanwhile, the catalase gene was significantly upregulated in female flies treated with the extracts. These enzymes play a crucial role in the inactivation of superoxide and hydrogen peroxide. The upregulation of antioxidant genes, such as *SOD1*, using quail egg oil extracts confirms the antioxidant effects of the bioactive compounds present in this extract.

There was a significant increase in the expression of the *CAT* gene in female *Drosophila* flies compared to what was observed in males. Singh et al. [76] reported sex-biased gene expression (SBGE), where the sexes differ in the amount a gene is expressed in *Drosophila* due to sexual dimorphism. SBGE represents sex differences in the abundance of all transcripts mapping to a given gene. In addition to dimorphism, in the total expression of a gene, the sexes may differ in other aspects of the transcriptome. It has also been stated that SBGE is associated with the chromatin state in *Drosophila melanogaster* and *Drosophila simulans* [77]. Furthermore, it has been suggested that sex-related characters evolve rapidly relative to non-sex-related characters in gene expression [78].

The *EIGER* gene in *Drosophila* is an inflammatory gene that is a homologous gene to TNF. It serves as a ligand of the TNF superfamily, triggering the JNK signaling pathway in *Drosophila*, which is a powerful genetic model to study the role of cell death in vivo and its physiological regulation [79,80]. Moreover, mRNA expression of the pro-inflammatory *UPD2* (IL-6) was downregulated via the QEYO. In mammals, injuries activate Toll-like receptor/NF-κB signaling in macrophages, which then express and secrete secondary cytostatic, pro-inflammatory cytokines. In *Drosophila*, distal puncture injury to the body causes the activation of Toll and Jun kinase (JNK) signaling via the hematopoietic system, and Toll and JNK signaling are coupled in their activation [81,82].

The effects of QEYO on these two important inflammatory cytokines further explain the anti-inflammatory effects of the bioactive compounds extracted with the various solvents. Based on our observations, we can tentatively postulate that the anti-inflammatory and antioxidant effects of QEYO result from the synergistic effect of these compounds present in interfering with the JNK signaling pathway. This interference significantly reduces the effects of pro-inflammatory mediators, partly due to the powerful free radical scavenging activities of these compounds.

## 5. Conclusions

In this study, compounds extracted from QEYO demonstrated a broad and diverse range of bioactive activities, including antioxidant, anti-inflammatory, antimicrobial, analgesic, anti-eczematic, and psoriatic effects. These pharmacological effects stem from the diverse array of compounds isolated using two distinct extraction solvents. Importantly, our research highlights the ability of these bioactive compounds from quail egg oil to modulate the regulation of anti-inflammatory genes, showcasing robust antioxidative effects. The observed effects can be attributed to the targeted regulation of specific genes, providing valuable insights into the potential therapeutic applications of QEYO-derived compounds.

## Figures and Tables

**Figure 1 antioxidants-13-00075-f001:**
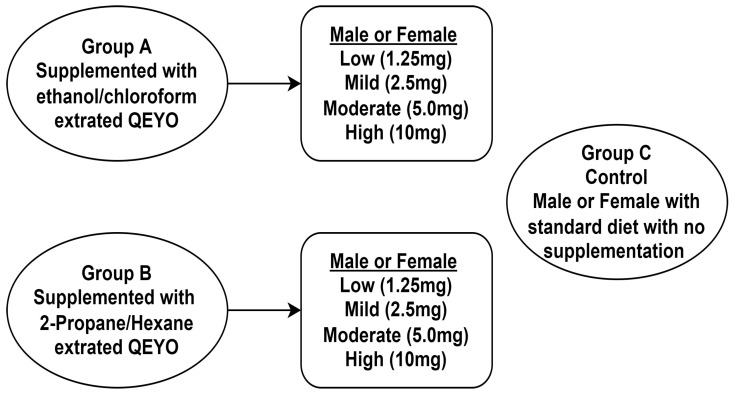
Experimental design.

**Figure 2 antioxidants-13-00075-f002:**
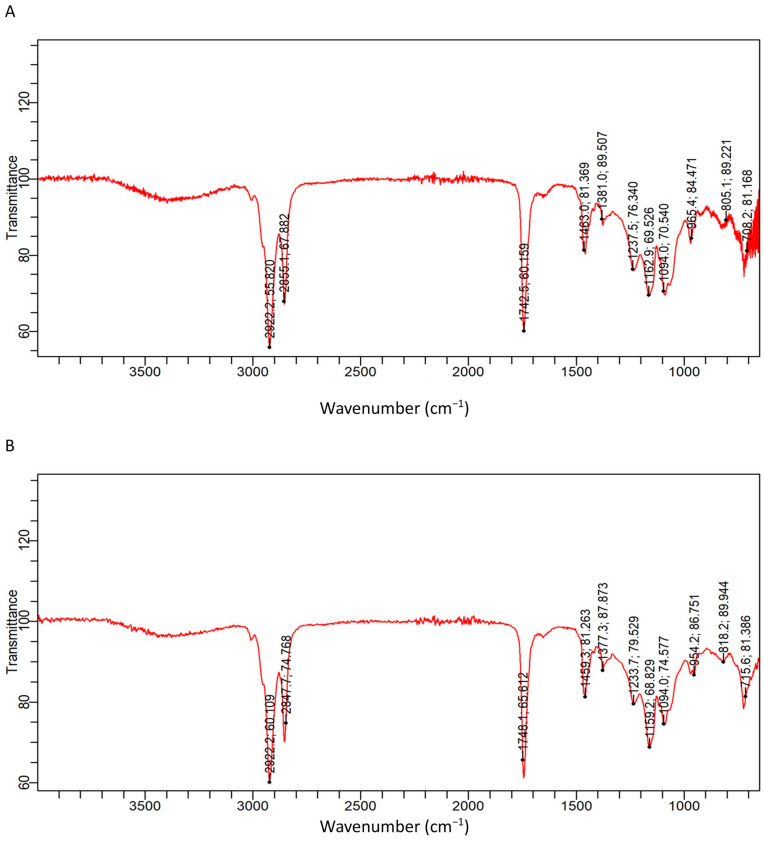
FTIR spectrum of the ethanol/chloroform (**A**) and 2-propanol/hexane (**B**) quail egg yolk oil extracts.

**Figure 3 antioxidants-13-00075-f003:**
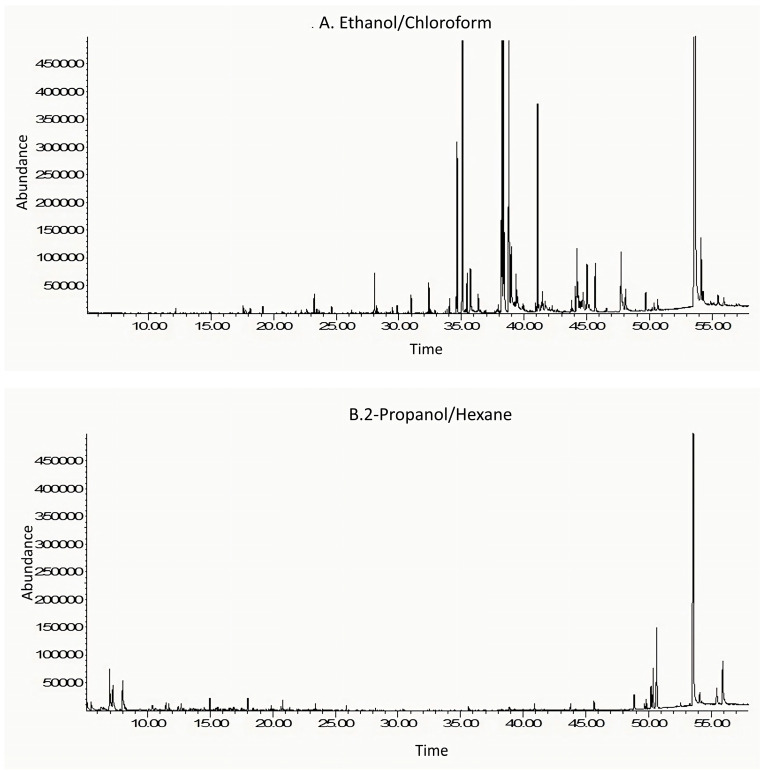
GC–MS spectrum analysis of the ethanol/chloroform (**A**) and 2-propanol/hexane (**B**) quail egg yolk oil extracts.

**Figure 4 antioxidants-13-00075-f004:**
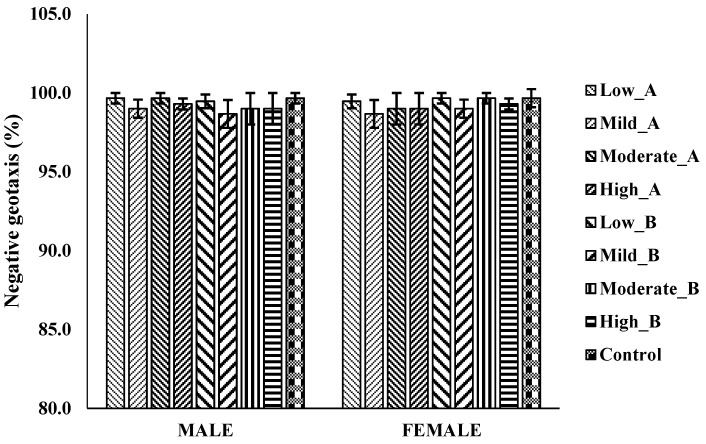
Negative geotaxis assay to evaluate locomotion in *D. melanogaster* flies treated with 2.5 mg/g (low); 5 mg/g (moderate); and 10 mg/kg (high) doses of A: ethanol/chloroform extract and B: 2-propanol/hexane extract of QEYO. The values have been given as the mean ± SEM (n = 3). These data were analyzed using the one-way ANOVA statistical test followed by Tukey’s post-hoc test using IBM SPSS v20.

**Figure 5 antioxidants-13-00075-f005:**
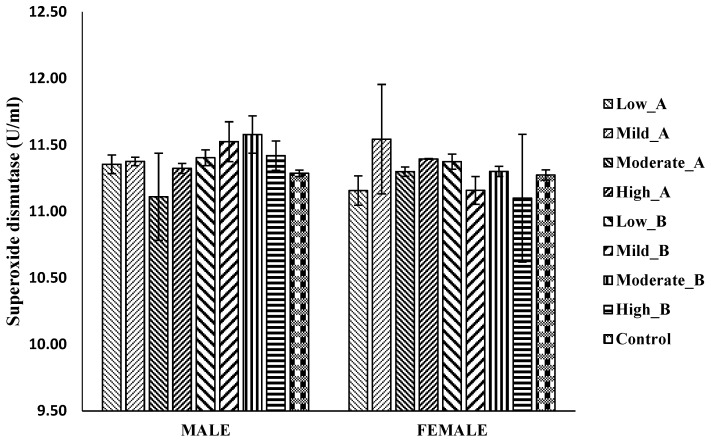
Levels of superoxidase dismutase (SOD) in *D. melanogaster* flies treated with low: 1.25 mg/g; mild: 2.5 mg/g; moderate: 5 mg/g; and high: 10 mg/g doses of A: ethanol/chloroform extract and B: 2-propanol/hexane extract of QEYO. The values have been expressed as the mean ± SEM (n = 3). These data were analyzed using the one-way ANOVA statistical test followed by Tukey’s post-hoc test using IBM SPSS v20.

**Figure 6 antioxidants-13-00075-f006:**
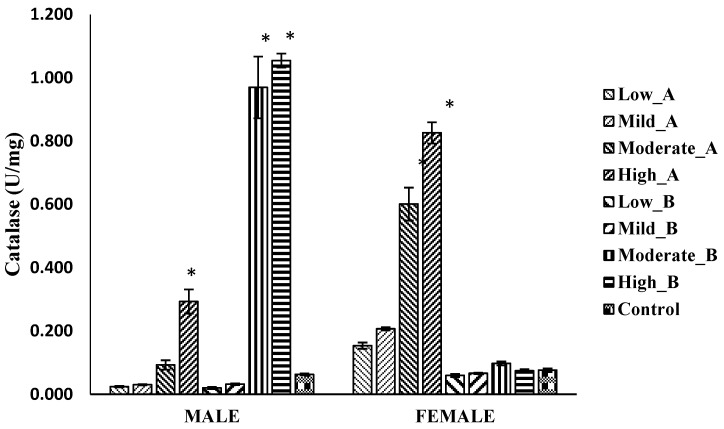
Catalase levels of *D. melanogaster* flies treated with low (1.25 mg/g), mild (2.5 mg/g), moderate (5 mg/g), and high (10 mg/g) doses of A: ethanol/chloroform extract and B: 2-propanol/hexane extract of quail egg yolk oil (QEYO). Values are expressed as the mean ± SEM (n = 3). These data were analyzed with the one-way ANOVA followed by Tukey’s post-hoc test using IBM SPSS v20. Bars with asterisks indicate significant differences from the control at *p* < 0.05.

**Figure 7 antioxidants-13-00075-f007:**
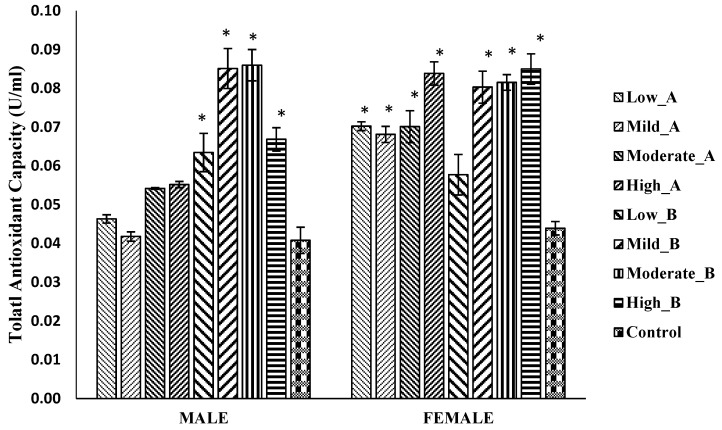
Total antioxidant capacity in *D. melanogaster* flies treated with low (1.25 mg/g), mild (2.5 mg/g), moderate (5 mg/g), and high (10 mg/g) doses of A: ethanol/chloroform extract and B: 2-propanol/hexane extract of quail egg yolk oil (QEYO). Values are expressed as the mean ± SEM (n = 3). These data were analyzed using the one-way ANOVA followed by Tukey’s post-hoc test with IBM SPSS v20. Bars with asterisks indicate significant differences from the control at *p* < 0.05.

**Figure 8 antioxidants-13-00075-f008:**
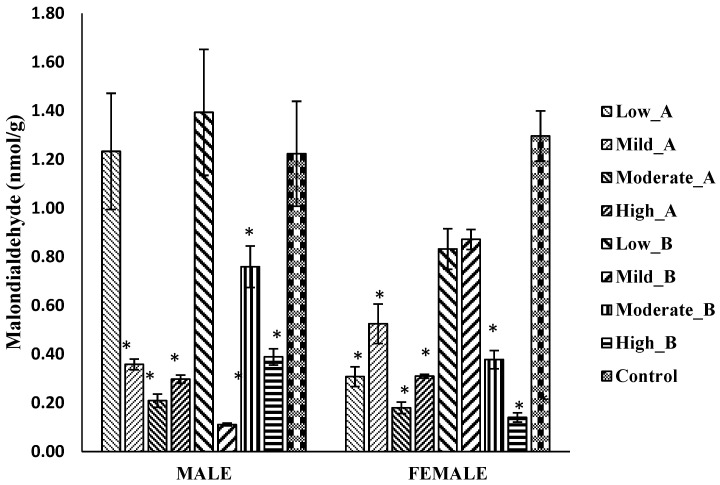
Malondialdehyde levels in *D. melanogaster* flies treated with different doses (low: 1.25 mg/g; mild: 2.5 mg/g; moderate: 5 mg/g; and high: 10 mg/g) of A: ethanol/chloroform extract and B: 2-propanol/hexane extract of quail egg yolk oil (QEYO). Values are expressed as the mean ± SEM (n = 3). These data were analyzed using the one-way ANOVA followed by Tukey’s post-hoc test with IBM SPSS v20. Bars with asterisks indicate a significant difference from the control at *p* < 0.05.

**Figure 9 antioxidants-13-00075-f009:**
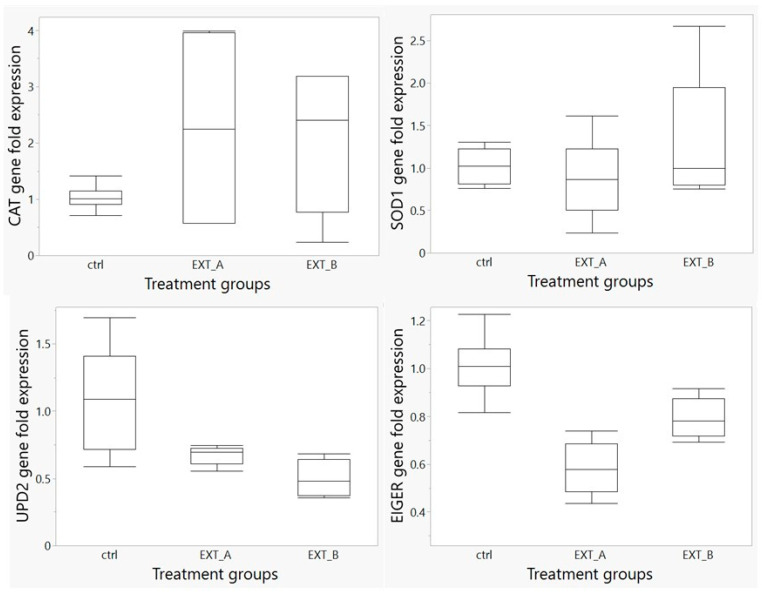
Gene expression analysis of the *CAT, SOD1, UPD2, and EIGER* genes in *D. melanogaster* flies treated with quail egg yolk oil (QEYO) extracts. The mean ± SEM fold change in the gene expression of the *CAT, SOD1, UPD2*, and *EIGER* genes in *D. melanogaster* flies treated with two different extracts of quail egg yolk oil (QEYO), namely EXT_A (ethanol/chloroform) and EXT_B (2-propanol/hexane), compared to the control (ctrl) non-treated group (n = 6) using a moderate dose of 5 mg/g. The gene expression levels of *EIGER and UPD2* showed a significant decrease (*p* < 0.05), while no significant change was observed in the expression of the *CAT* and *SOD1* genes (*p* > 0.05).

**Figure 10 antioxidants-13-00075-f010:**
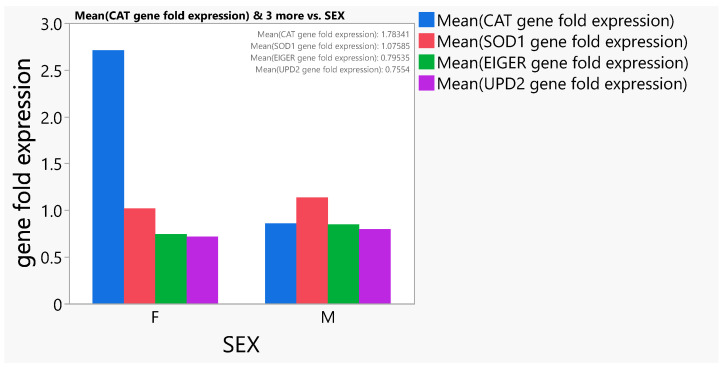
Means of the gene expression levels of *CAT, SOD1, EIGER, and UPD2* of *D. melanogaster* flies of both sexes.

**Table 1 antioxidants-13-00075-t001:** The *Drosophila melanogaster* gene name, annotation symbol, fly base ID, accession number, and primer sequences were used.

Gene	Annotation Symbol	Fly Base ID	Accession	Forward (5′ to 3′)	Reverse (5′ to 3′)
*SOD1*	CG11793	FBgn0003462	NM_057387.5	CGGTCACACCATAGAAGATACC	CAGACAGCTTTAACCACCATTTC
*CAT*	CG6871	FBgn0000261	NM_080483.3	TGGTCGTCTGTTCTCCTACT	CCGCTGGAAGTTCTCAATCT
*UPD2*	CG5988	FBgn0030904	NM_001370039.1	TGAGGCAACTTCCAAAGAGAG	CGGATCTGGCTGAAAGAAGAG
*EIGER*	CG12919	FBgn0033483	NM_165735.4	TTGACCATAAACGCCTCCTATC	GTGAAAGTTGAGACGCTCCT
*RPL32*	CG7939	FBgn0002626	NM_170460.2	GTCGTCGCTTTGTCATCT	GCAGGTTGTAGCCCTTCTT

*SOD1*: superoxide dismutase 1, *CAT*: catalase, *UPD2*: unpaired 2, *EGR*: tissue necrotic factor Eiger, and *RPL-32* = 60S ribosomal protein large subunit-32.

## Data Availability

Data are contained within this article.

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
