# Peer review of "Antioxidant and Anti-Inflammatory Properties of Quail Yolk Oil via Upregulation of Superoxide Dismutase 1 and Catalase Genes and Downregulation of EIGER and Unpaired 2 Genes in a *D. melanogaster* Model"

_antioxidants, 2024, doi:10.3390/antiox13010075_

Round 1

Reviewer 1 Report (Previous Reviewer 1)

Comments and Suggestions for Authors

- All the data were analyzed with one-way Anova which is only appropriate if the data are parametric. It is unlikely that that is the case for the expression or activity studies. 

Thank you so much for the observations, the data was analyzed using one-way ANOVA comparing the treated groups (A and B) with the control group

That the data was analyzed by one-way Anova was already described. The concern was whether that is the right test. To determine that, they have to test for normal distribution which they don't mention in their response. 

Author Response

- All the data were analyzed with one-way Anova which is only appropriate if the data are parametric. It is unlikely that that is the case for the expression or activity studies. 

Thank you so much for the observations, the data was analyzed using one-way ANOVA comparing the treated groups (A and B) with the control group

That the data was analyzed by one-way Anova was already described. The concern was whether that is the right test. To determine that, they have to test for normal distribution which they don't mention in their response. 

  • Thank you for the observation; A Test for normality on the data was performed and the data was found to be normally distributed before running the ANOVA.

Lines 225-230.

Reviewer 2 Report (Previous Reviewer 2)

Comments and Suggestions for Authors

Muhammad Sani Ismaila et al. analyze the different compounds in QEYO and investigate the antioxidant activity, and quantify the toxicity. This work is very interesting, but there are still some major points need focused.

1. In this study, the compounds extracted from QEYO were found to have broad and diverse bioactive activities ranging from antioxidant, anti-inflammatory, antimicrobial, nalgesic, anti-eczematic, and psoriatic activities, and these pharmacological effects are attributed to the various compounds isolated from 2 extraction solvents.The author should discuss the main substances that may play pharmacological effects and their mechanism.

2. The key figure is not clear. Eg Fig 2, 3, 9.

Author Response

Muhammad Sani Ismaila et al. analyze the different compounds in QEYO and investigate the antioxidant activity, and quantify the toxicity. This work is very interesting, but there are still some major points need focused.

  1. “In this study, the compounds extracted from QEYO were found to have broad and diverse bioactive activities ranging from antioxidant, anti-inflammatory, antimicrobial, nalgesic, anti-eczematic, and psoriatic activities, and these pharmacological effects are attributed to the various compounds isolated from 2 extraction solvents.”The author should discuss the main substances that may play pharmacological effects and their mechanism.
  • Thank you for the observation; This was addressed please see Lines 410-434
  1. The key figure is not clear. Eg Fig 2, 3, 9.

Thank you; The quality of the figures improved

Reviewer 3 Report (Previous Reviewer 3)

Comments and Suggestions for Authors

Some improvements are present in the re-submitted version of the paper. However, Discussion shows speculations and it should instead be based on the data obtained in the experiment. Conclusion should be also rewritten since peculations are present: the sentence (line 480-482) does not make any sense because the upregulation of SOD is not significant.

Minor:

-        Line 426: it is not Table 1;

-        Line 448: SOD and CAT are antioxidant enzymes and not pro-oxidants.

Comments on the Quality of English Language

English needs some improvement.

Author Response

Some improvements are present in the re-submitted version of the paper. However, Discussion shows speculations and it should instead be based on the data obtained in the experiment. Conclusion should be also rewritten since peculations are present: the sentence (line 480-482) does not make any sense because the upregulation of SOD is not significant.

  • Thank you for the observations; the Discussion was centered on the results obtained.
  • The conclusion was re-written. The sentence on lines 484-485 was deleted.

Minor:

-        Line 426: it is not Table 1;

  • Thank you: corrected to Table2.

-        Line 448: SOD and CAT are antioxidant enzymes and not pro-oxidants.

  • Thank you; This was corrected.

Comments on the Quality of English Language

English needs some improvement.

  • Language use improved

Round 2

Reviewer 1 Report (Previous Reviewer 1)

Comments and Suggestions for Authors

The authors have addressed my concerns

Author Response

Thank you for the comments.

Reviewer 2 Report (Previous Reviewer 2)

Comments and Suggestions for Authors

The authors cannot just detect some indicators and get the conclusions.

The author should discuss the main substances that may play pharmacological effects and their mechanism. 

Author Response

The authors cannot just detect some indicators and get the conclusions.

The author should discuss the main substances that may play pharmacological effects and their mechanism. 

  • Thank you so much for your comments: The bioactive compounds responsible for the pharmacological effects were further discussed in details with their mechanism of action.

Reviewer 3 Report (Previous Reviewer 3)

Comments and Suggestions for Authors

There have been some improvements in the Discussion section as requested.

Comments on the Quality of English Language

English needs to be carefully revised. Some mistakes are present and some sentences are to be shortened.

Author Response

There have been some improvements in the Discussion section as requested.

English needs to be carefully revised. Some mistakes are present and some sentences are to be shortened

  • Thank you so much for your comments: The manuscript was carefully revised, mistakes were amended and redundant sentences where deleted or rephrased.

Round 3

Reviewer 2 Report (Previous Reviewer 2)

Comments and Suggestions for Authors

 Accept in present form

This manuscript is a resubmission of an earlier submission. The following is a list of the peer review reports and author responses from that submission.

Round 1

Reviewer 1 Report

Comments and Suggestions for Authors

This manuscript aims to describe effects of quail egg yolk oil (QEYO)on oxidative stress and inflammatory processes in the Drosophila model system. Although describing such effects would be of interest, there are several major problems:

- The authors do not detect any biological effects due to no significant difference in the negative geotaxis assay. Therefore, changes in gene expression or protein activity are meaningless because they do not seem to have a functional consequence for the organism.

- The data in figure 9 are either not accurately described/presented or do not show what the authors claim. Shown is fold change, presumably to untreated controls, and if that is the case eiger expression is about 15fold increased in females and about the same in males. Similarly, catalase would actually be decrease in females and about the same in males. This is opposite of what the authors claim, namely that antioxidant genes are increased and inflammatory genes are decreased by QEYO. A significant difference between treated males and females does not show an effect of treatment

- In figure 8, it is surprising that high A and moderate B is not significant because they appear even lower than some of the other bars when compared to the control

- In general, the data show dramatic differences between males and females but this is not discussed

- The authors claim that they did the activity measurements in haemolymph but the method they described does not result in the supernatant only containing haemolymph

- w1118 is a mutant that has specific phenotypes and it is NOT wild type as the authors claim

- All the data were analyzed with one-way Anova which is only appropriate if the data are parametric. It is unlikely that that is the case for the expression or activity studies

- There is no n given in any of the figures, although were the data from technical repeats or biorepeats? 

Reviewer 2 Report

Comments and Suggestions for Authors

Muhammad Sani Ismaila investigated the Antioxidant and anti-inflammatory properties of quail yolk oil. This work is very interesting, but there are still some major points need focused.

1. Which dose of quail yolk oil show best effect ? Equivalent human dose?

2. It is advised to characterize the antioxidant activity of the quail yolk oil.

3. Studies of antioxidant mechanisms are more convincing at the protein level.

Reviewer 3 Report

Comments and Suggestions for Authors

Discussion and conclusions are not supported by data. Changes in the composition of the quail yolk oil are not reported and discussed. The Ms. lacks of calculation of the concentrations of  active compounds. Are fatty acids endowed of  antioxidant activity? Which is most effective compounds? Which is the relationship among bioactive compounds and antioxidant and anti-inflammatory properties of different samples? It lacks the rationale of the work.

Comments on the Quality of English Language

It needs little revision